# C-Reactive Protein as Predictor for Infectious Complications after Robotic and Open Esophagectomies

**DOI:** 10.3390/jcm11195654

**Published:** 2022-09-26

**Authors:** Florian Richter, Anne-Sophie Mehdorn, Thorben Fedders, Benedikt Reichert, Jan-Hendrik Egberts, Thomas Becker, Julius Pochhammer

**Affiliations:** 1Department of General, Visceral, Thoracic, Transplantation and Pediatric Surgery, University Hospital Schleswig-Holstein Campus Kiel, Arnold-Heller-Str. 3, Building C, 24105 Kiel, Germany; 2Department of Surgery, Israelit Hospital, Orchideenstieg 14, 22297 Hamburg, Germany

**Keywords:** RAMIE, robotic surgery, robotic esophagectomy, minimally invasive surgery, esophageal cancer, esophagus, Ivor-Lewis, complications, PIC, postoperative infectious complications, CRP, C-reactive protein

## Abstract

Introduction: The value of C-reactive protein (CRP) as a predictor of anastomotic leakage (AL) after esophagectomy has been addressed by numerous studies. Despite its increasing application, robotic esophagectomy (RAMIE) has not been considered separately yet in this context. We, therefore, aimed to evaluate the predictive value of CRP in RAMIE. Material and Methods: Patients undergoing RAMIE or completely open esophagectomy (OE) at our University Center were included. Clinical data, CRP- and Procalcitonin (PCT)-values were retrieved from a prospectively maintained database and evaluated for their predictive value for subsequent postoperative infectious complications (PIC) (AL, gastric conduit leakage or necrosis, pneumonia, empyema). Results: Three hundred and five patients (RAMIE: 160, OE: 145) were analyzed. PIC were noted in 91 patients on postoperative day (POD) 10 and 123 patients on POD 30, respectively. Median POD of diagnosis of PIC was POD 8. Post-operative CRP-values in the robotic-group peaked one and two days later, respectively, and converged from POD 5 onward compared to the open-group. In the group with PIC, CRP-levels in the robotic-group were initially lower and started to differ significantly from POD 3 onward. In the open-group, increases were already noticed from POD 3 on. Procalcitonin levels did not differ. Best Receiver operating curve (ROC)-results were on POD 4, highest negative predictive values at POD 5 (RAMIE) and POD 4 (OE) with cut-off values of 70 mg/L and 88.3 mg/L, respectively. Conclusion: Post-operative CRP is a good negative predictor for PIC, after both RAMIE and OE. After RAMIE, CRP peaks later with a lower cut-off value.

## 1. Introduction

Esophagectomies remain among the most challenging upper gastrointestinal procedures. Esophageal cancer (EC) is the main indication for esophagectomy, but it is also performed for benign conditions, i.e., achalasia, Boerhaave syndrome, or iatrogenic injuries [1]. Despite continuous surgical innovations, esophagectomy remains a highly complex procedure that is accompanied by severe intra- and postoperative complications. After the introduction of minimally invasive and hybrid minimally invasive esophagectomies (MIE) and (HMIE), respectively, robot-assisted minimally invasive esophagectomy (RAMIE) has become an alternative to the conventional open esophagectomy (OE) [2,3]. RAMIE entails less surgical trauma and more precise movements without compromising the oncological quality of the procedures [4]. RAMIE is proven to significantly influence the health-related quality of life (HRQoL) and long-term survival of the patients post esophagectomy [5]. Robotic procedures have as well found their way into many other surgical fields, i.e., gynecological, hepatic, colorectal or reconstruction surgery [6,7,8,9,10,11].

Although recent years have seen a decline in the morbidity rates post esophagectomies, rates of postoperative infectious complications (PIC) and mortality are still up to 40% [12]. PIC encompasses pulmonary and gastric conduit-related complications. Pulmonary complications include pneumonia with an incidence of 20–40%, pleural effusion, and pleural empyema [13]. Gastric conduit-related complications mainly comprise perforation, bleeding, ischemic changes, and anastomotic leakages (AL) in and of the gastric conduit. AL is defined as a defect of the intestinal wall at the anastomotic site with interactions between intra- and extra-luminal compartments [14]. AL secondary to esophagectomy has an incidence of 5–30% [15,16], and its associated morbidity and mortality have an incidence of 20–50% [17,18]. However, the most feared postoperative complication is mediastinitis, followed by septicemia. Mediastinitis-associated mortality rates of up to 80% have been reported [19].

Hence, timely and appropriate diagnosis and treatment of PIC is crucial [20]. Yet, both surgical trauma and PIC cause a systemic inflammatory infection, resulting in an increase in white blood cell (WBC) count, C-reactive protein (CRP), and procalcitonin (PCT) within the first postoperative day (POD) [21]. However, CRP and other inflammatory markers may not exactly determine the origin of the systemic inflammatory reaction. In the absence of any PIC, the infectious values reach their maximum on the second or third POD and normalize in the course of the first postoperative week [22]. Hence, WBC-count and the level of are crucial for the early detection of PIC in addition to clinical symptoms. In the presence of PIC, the infection levels are usually initially higher and do not significantly drop during the postoperative course [23]. Attempts have been made to correlate the predictability of PIC with CRP- and PCT-serum levels [24]. In the course of the Enhanced Recovery After Surgery (ERAS)-program, CRP cut-off values are already taken into account when planning the early discharge of patients after major abdominal or colorectal surgery [25,26]. However, no exact cut-off values have been determined yet.

To our knowledge, only Babic et al. consider CRP as a possible complication marker in MIE; yet without differentiating between MIE, HMIE and RAMIE, so far [20]. However, they utilized different definitions for endpoints and calculated diverging cut-off values; therefore, a universally valid cut-off value has not yet been identified.

Therefore, the present study aims to evaluate the positive and negative predictive value of perioperative CRP- and PCT-levels to predict PIC within 30 days after RAMIE, compared to OE.

## 2. Materials and Methods

### 2.1. Patient Cohort

All patients underwent surgical resection in the Department of General, Abdominal, Transplantation, Thoracic and Pediatric Surgery, University Hospital Schleswig Holstein (UKSH), Campus Kiel between December 2008 and January 2021. Patients with and without an underlying malignant disease of the esophagus were included. Patient data were collected in a prospectively maintained database. In total, 406 patients were included for this study. All patients gave written informed consent before inclusion in the study. Clinical data were obtained from patient files and the clinical research database of the oncological biobank, BMB-CCC, of the Medical Faculty of the University of Kiel. The study was approved by the local ethics committee of the Medical Faculty, Kiel University (reference no. A110/99), according to the principle of Helsiniki. Only de-identified data were used for further analysis.

### 2.2. Treatment Protocol

The operations were performed in a standardized manner [2,27,28,29,30]. Post-operative care consisted of transfer to the intensive care unit, monitoring and collection of daily blood samples, especially the analysis of CRP- (mg/L) and PCT-levels (µg/L). Classification of the pathological tumor stage was conducted at the Institute of Pathology, UKSH Campus Kiel, according to the edition of the TNM-classification [31].

### 2.3. Endpoints

The primary endpoints were PIC and CRP- as well as PCT-values. PIC included both pulmonary infectious complications (IC), i.e., pneumonia, pleural effusion, and pleural empyema, as well as AL, i.e., insufficiency of the gastro-esophageal anastomosis or gastric conduit, including ischemia of the upper end of the gastric conduit adjacent to the anastomosis, and necrosis of the gastric conduit or anastomosis. Pneumonia was diagnosed by the presence of pulmonary infiltrate on chest X-ray together with elevated CRP levels and/or clinical signs. Pleural empyema was diagnosed by means of computed tomography and subsequent thoracentesis. The definition of an AL ranged from a small fistula to a complete rupture of the esophagogastrostomy. All AL were detected through endoscopy, supported by computed tomography, if necessary. For further evaluation, PIC were differentiated into surgical (sPIC) and nonsurgical (nsPIC). The former included leakage and necrosis of the gastric tube, the latter pulmonary complications.

### 2.4. Statistical Analysis

The database was managed in MS Access (Redmond, WA, USA), and JMP 15.1 software (SAS Institute Inc., Cary, NC, USA) was used for statistical analysis. The baseline data and occurrence of endpoints were analyzed using the parametric Students’ *t*-test (normally distributed data) or the nonparametric Mann-Whitney-Wilcoxon test for continuous variables (non-normally distributed). The Chi-Square test or Fisher’s exact test were used for categorical variables wherever appropriate. CRP and PCT data are presented in their original unit of measurement. Data distribution was explored, and positively skewed score data underwent log-transformation prior to formal analysis. To compare the values between patients in PIC- and non-PIC-groups, the unpaired (Two Sample) *t*-test was performed on the data that underwent log-transformation.

Univariate analysis of the groups was performed by binary logistic regression. Multivariable analysis for the primary endpoints incorporated all independent variables, including both significant and insignificant data from the univariate analysis, and consisted of multiple regression analysis with backward elimination. The procedure was rerun until no explanatory variable was left that could be removed without markedly worsening the prediction of the endpoint. The optimal cut-off value of CRP was calculated by using the receiver operating characteristics (ROC) analysis, which plots true positives (sensitivity) against false positives (1−specificity), and Youden’s J Statistic, which uses the maximum vertical distance of the ROC curve from the point on the diagonal line (chance line); it maximizes the difference between true positives and false positives. All reported *p*-values were 2-sided, and a *p*-value < 0.05 was considered statistically significant.

## 3. Results

The final analysis of predictive quality included 160 patients (52.5%) who underwent RAMIE and 145 patients (47.5%) who underwent OE (Table 1, Figure 1). There were no baseline demographic differences between the two groups (RAMIE vs. OE). Besides a trend toward adenocarcinoma among RAMIE-patients, they were more often treated with an Ivor-Lewis procedure and received more neoadjuvant therapy. Procedures in the RAMIE-group were shorter, while more transhiatal resections were performed in the OE-group. Yet, the incidence of PIC did not differ between the groups. In total, 91 PIC (29.8%) were noticed until POD 10 and 123 PIC (40.3%) until POD 30 (Table A1). The median of detection was on POD 9 (1-30) and POD 7 (1-32) for the two groups, respectively (*p* = 0.81). The most common PIC was AL, which was found in 49 (30.6%) and 41 (28.3%) patients after RAMIE and OE, respectively (*p* = 0.65). Pneumonia was diagnosed in 72 patients (23.6%) during the postoperative treatment without significant difference between the groups (RAMIE vs. OE, *n* = 35 vs. *n* = 37) (Table A1). The median POD of diagnosis of pneumonia was POD 4 (0-30) and POD 6 (0-18) in RAMIE- and OE-patients, respectively. Necrosis of the gastric conduit occurred in 9 (3.0%) patients, and leakage of the gastric conduit in 10 (3.3%) patients without significant difference between both groups. Surgical site infections (Grade A and B after Classification of Centers for Disease Control and Prevention (CDC)) were found in 3 (1.9%) and 12 (8.3%) RAMIE- and OE- patients, respectively (*p* < 0.01).

In the next step, groups were further stratified by the development of PIC (PIC vs. NoPIC) to identify risk factors (Table 2). No significant differences between the groups were found for gender, age, ASA, smoking, and entity, while BMI was slightly higher in PIC-patients. No differences in the procedure and surgical approach were found. The same accounts were obtained for neoadjuvant treatment and adjuvant radiotherapy. Of note, patients with PIC had a longer hospital stay and received significantly less adjuvant chemotherapy.

Considering the postoperative CRP-levels of patients included, they differed significantly from POD 2 onward for patients with and without PIC (Table 3). The increase in values from POD 1 was also steeper with a later peak in the PIC-group. Among RAMIE-patients, the increase of CRP in the case of PIC-development was slower; values only differed from POD 3 onward (Figure 2). In the OE-group, on the other hand, in case of PIC-development, CRP-levels were significantly elevated already from POD 2 onward compared to noPIC-development. Post-operative PCT values did not differ significantly between the groups though and were not further taken into consideration.

Within the PIC-group, CRP-values in RAMIE-patients were significantly lower at POD 1 and 2, but higher at POD 4, compared to that of OE-patients due to a slower increase and a later peak after RAMIE, yet without being significantly different (Figure 3). No difference existed on POD 3.

RAMIE-patients without PIC showed significantly lower CRP-values at POD 1, compared to OE-patients. CRP-levels increased slower and were significantly higher at POD 3 and 4. From POD 5 onward, CRP-values no longer differed between both approaches.

In order to determine test quality, predictive values, and best cut-off, receiver operating characteristic (ROC) analysis was performed. The best ROC results were on POD 4, with an area under the curve (AUC) 0.73 and 0.74 for RAMIE and OE, respectively. The highest negative predictive values were >80% at POD 5 (RAMIE) and POD 4 (OE). The best cut-off values were 70.0 mg/L and 88.3 mg/L for RAMIE and OE, respectively (Table 4). The best positive predictive values for identifying later PIC were found at POD 4 and POD 5, with cut-off values of 168 mg/L and 142 mg/L for RAMIE and OE, respectively (Table 4).

In addition, we compared the time of onset of surgical (sPIC) and nonsurgical PIC (nsPIC) and the level of CRP at POD 4 for the different approaches.

NsPIC occurred in median at POD 4 (0-28) after RAMIE, and at POD 5.5 (0-18) after OE (*p* = 0.55). CRP-values at POD 4 were 155.4 + 67.5 mg/L and 140.1 + 75.1 mg/L after RAMIE and OE, respectively (*p* = 0.71).

sPIC occurred in median at POD 9 (1-30) and POD 7 (1-32) and CRP values at POD 4 were 193.3 + 85.6 mg/L and 161.0 + 84.9 mg/L (*p* = 0.09), after RAMIE and OE, respectively. For both, RAMIE and OE, the differences in occurrence between sPIC and nsPIC as well as CRP levels at POD 4 were not statistically significant (*p* = 0.37 and *p* = 0.12).

## 4. Discussion

In this single-center study, we compared postoperative courses after RAMIE and OE regarding PIC and reported on the predictability of postoperative CRP- and PCT-levels. We found that CRP-levels after RAMIE showed a delayed increase and peak, compared to OE. Further, we found this delay to be independent of the occurrence of PIC. In the case of PIC-development, postoperative CRP-levels differed from POD 2 onward, compared to patients without PIC, regardless of the type of procedure. We attributed the later CRP-peak after RAMIE to different CRP-kinetics after RAMIE. The best positive predictive value for the development of PIC in RAMIE-patients was a cut-off value of 168 mg/L for CRP on POD 4 (compared to 142 mg/L after OE), while the best negative predictive value for the development of PIC after RAMIE was a CRP-value of 70 mg/L on POD 5, compared to 88.3 mg/L after OE for EC. However, we did not find any prognostic value for postoperative PCT-levels.

Previously, it has been shown that early detection of PIC is essential not only for the treatment of complications but also for patient survival [20,32,33,34,35,36,37,38]. PICs, according to the definition used for this paper, include surgical site infections, pneumonia, pleural empyema, leakage or necrosis of the gastric conduit, and anastomotic leakage [33,39]. Indisputably, these complications are of different importance and severity. Especially necrosis and leakage of the gastric conduit, as well as anastomosis, may be life-threatening or may at least prolong the postoperative reconvalescence. Previously, these complications caused mortality rates of up to 80% [19]. Often, these PIC could only be treated by reoperation, leading to salivary fistula, accompanied by reduced postoperative HRQoL [40]. Current strategies for the treatment of PIC, especially necrosis of the gastric conduit and AL are less invasive and include placements of esophageal stents and endoscopic vacuum closure systems, which render less harm to the patient and promote quicker recovery and an elevated HRQoL [41]. Yet, PIC does not only consist of necrosis or leakages of the gastric tube, but also pulmonary complications, especially pneumonia and pleural empyema [42,43,44]. In particular, patients with a previous history of smoking seem to be more susceptible to postoperative pulmonary complications [42]. It was postulated that less invasive and shorter procedures, especially for the thoracic part of the procedure, would cause fewer pulmonary infectious complications [34,43,44,45,46]. Glatz et al. found HMIE to cause lesser pulmonary and overall complications, compared to OE, without taking infectious parameters into consideration [47]. In our analysis, however, we found similar incidences of PIC in both cohorts (40.0% and 40.7%, respectively, ns). The overall incidence of pneumonia was 23.6% in both cohorts with slightly more pneumonia detected after OE. Incidence of AL was low in the whole cohort (29.6%). We, hence, cannot support the previously mentioned results of less infectious complications after MIE by our experience.

PIC are either diagnosed by clinical signs or by elevated infectious parameters. Yet, diagnosis of a tumor itself is often accompanied by elevated infectious parameters, and systemic inflammatory responses seem to influence the tumor’s growth, progression, and prognosis [48,49]. Surgery and preparation during surgery themselves cause trauma, influence infectious parameters and potentially tumor growth [50,51]. Within the development of surgical procedures, minimally invasive procedures have been developed and said to be causing less surgical trauma [20]. The more historic procedures are said to cause more surgical trauma and, therefore, increase inflammatory reactions, thus causing higher blood levels of inflammatory markers. In a case-controlled study, Scarpa et al. identified HMIE to accompany less systemic inflammatory responses and reduced surgical trauma using identifiably lower postoperative CRP-levels post HMIE [52]; they further found lower CRP-levels on POD 1 in HMIE-patients. RAMIE or hybridRAMIE are considered to be even less invasive and less harmful with regard to the surgical approach. Additonally, minimally invasive approaches do not only lead to a better cosmesis, but also to a smaller surgical trauma and, thereby, quicker recovery and better HRQoL [4]. However, smaller surgical trauma needs to be seen relatively as the inner trauma made by surgical preparation needs to be taken into consideration as well. In a clinical setting, this must be kept in mind when evaluating the postoperative infectious parameters, i.e., CRP- and PCT-levels. Usually, postoperative infectious parameters reach their maximum on POD 2 [21]. In this context, CRP is usually said to be less accurate than PCT [53,54]. While we found a good positive predictability of postoperative CRP-levels on POD 4 for the development of PIC after RAMIE, we could neither identify any difference in postoperative PCT-values nor positive or negative predictability of PCT-levels. In a recent study, Yang et al. identified both CRP and PCT to be valuable negative predictors for early monitoring (POD 3) of PIC after laparoscopic colorectal surgery [54]. Chen et al. identified Interleukin 6 to have high prognostic values for postoperative pneumonia, while only identifying moderate prognostic value for WBC, CRP and PCT [42]. In line with our results are the results by Babic et al.: Firstly, they reported an earlier increase in postoperative CRP-levels after OE, compared to HMIE, MIE, or RAMIE [20]. Secondly, they reported a higher increase in postoperative inflammatory markers after OE, compared to any type of minimally invasive esophagectomy on POD 1 and 2. As expected, patients with uncomplicated courses presented with lower CRP-values in both the OE- and MIE-groups. Babic et al. further identified CRP-levels > 200 mg/L on POD 2 to be a positive and an independent predictor for PIC, including pulmonary complications and AL, especially after MIE [20]. In our analysis, we found CRP-values > 168 mg/dl on POD 4 and >142 md/dl on POD 5 for RAMIE and OE, respectively, to have a high positive predictive value for the development of a PIC. Neary et al. found a CRP-level of 218 mg/dl on POD 2 to come along with a negative predictability of 97% for the development of an AL [55]. Chen et al. report the CRP-levels in POD 3 to have the highest diagnostic value [42]. Barrie et al., on the other hand, identified an even later CRP-peak (POD 7) to be predictive of the development of delayed complications [56].

Non-surgical complications may also seriously impact reconvalescence. The intention of our study was to calculate cut-off values to predict an uncomplicated course. Therefore, all PIC, sPIC as well as nsPIC, were combined. It is possible that a different steep and high increase of CRP in sPIC and nsPIC influences the predictive power. However, no significant difference could be found in our study for both forms of PIC with respect to time of onset as well as for the level of CRP at POD4.

Regarding postoperative PCT-levels, Booka et al. conducted a study, analyzing the association between postoperative PCT-levels, PIC and oncological outcome; they identified a prognostic value of 100% of PCT on POD 7 for development of PIC and overall outcome. However, similar to our analysis, they did not find a difference on POD 2 [57]. Hence, it may be concluded, that later PCT-values are of greater information compared to early PCT-values. However, we only identified PCT-levels until POD 5.

In order to meet the inaccuracy and limits of retrospective, postoperative data, different prediction models have been developed to foresee the probability of PIC or an AL [36,43,58,59]. Most prediction models are based on CRP, WBC, CRP-WBC-ratios, CRP-neutrophiles-ratios and albumin [55,56,59]. Our study is also based on lab results that are in line with most predictive studies [36,58]. In a previous study from our center, analyzing the predictability of infectious parameters on anastomotic leakages post bariatric surgery, we found WBC to be of little use in the identification of AL post bariatric surgery, and we, therefore, excluded it from the analysis of this paper and focused on CPR and PCT [60]. However, the novelty of our study is that its predictability is fully based on homogenous cohorts (full RAMIE vs. full OE), compared to others who included open, HMIE, and MIE in the analysis [20,36,59]. Nevertheless, to provide a solid prediction score, the Noble/Underwood (NUn) score was developed by Noble and Underwood [58,59]. The NUn score includes WBC, CRP and albumin on POD 4 [61]. Using the NUn-score, Bundred et al. found the lowest predictability on POD 2 and the highest on POD 6 [58]; they also identified a good NUn score-based predictability for major complications on POD 4 with CRP-values of 219 ± 95 mg/L. Findlay et al. had previously reported on a predictive CRP-value of 156 mg/L using the NUn-score, focusing on results on POD 4 [62]. Liesenfeld et al. estimate CRP-levels below 155 mg/dl between POD 3 and 7 to have a high negative prediction for not developing a postoperative AL using the NUn-score [61]. However, according to our experience, it is of great importance and interest to detect postoperative complications as early as possible in order to prevent further complications and associated morbidity.

## 5. Limitations

The strength of this study lies in its strict inclusion of only full RAMIE and full OE, thereby providing distinct cohorts. Yet, it has to deal with the limitations of a small collective, and thus, limited statistical power and retrospective data acquisition. Also, we have not yet tested the cut-off values calculated in our cohort in a validation cohort, as recommended for predictive models. Further, we only provide data for CRP and PCT without taking other inflammatory markers, i.e., WBC, IL-6 or also cytokine levels in drainage secretion, into consideration. It is expected that in the future machine learning will be able to incorporate significantly more clinical factors, such as movement behavior and respiratory rates, into regression models and thus achieve more accurate predictions.

## 6. Conclusions

RAMIE cause less surgical trauma. Consequently, postoperative CRP-values increase slower and show a later peak after RAMIE, compared to OE. However, CRP-levels on POD 5 seem to have a good negative predictability at values under 70 mg/L. This should be considered when evaluating postoperative laboratory values to achieve better predictive values. For example, a cut-off value of 70 mg/L of POD 5 results in a NPV of 83.7% to rule out infectious complications after RAMIE.

## Figures and Tables

**Figure 1 jcm-11-05654-f001:**
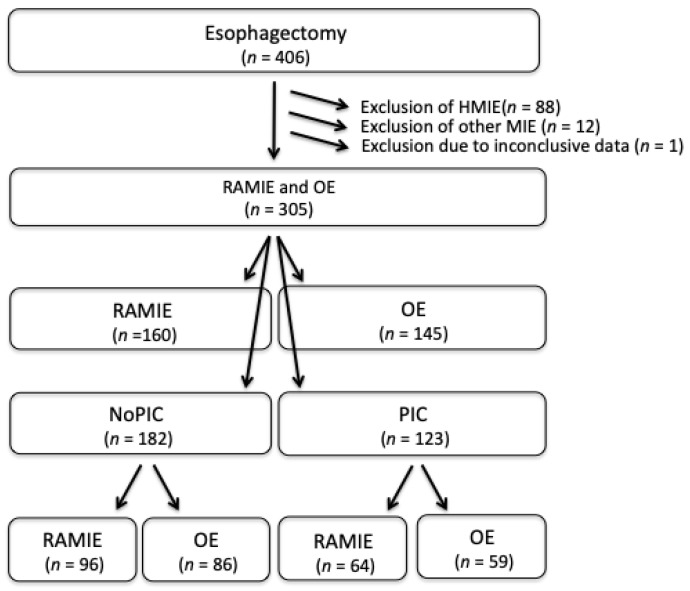
Flow-Chart of the inclusion of the study. HMIE: Hybrid minimally invasive esophagectomy, MIE: minimally invasive esophagectomy, NonPic: non-postoperative infectious complications, PIC: postoperative infectious complications, RAMIE: Robot-assisted minimally invasive esophagectomy, OE: open esophagectomy.

**Figure 2 jcm-11-05654-f002:**
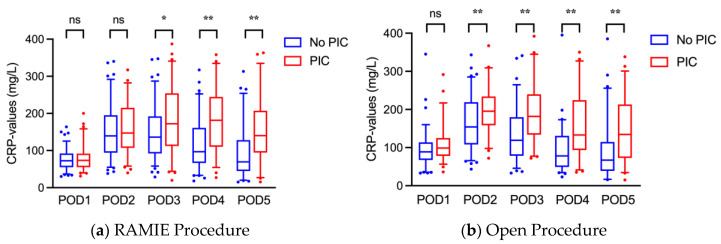
Postoperative CRP-values of patients undergoing RAMIE (*n* = 160) (**a**) or OE (*n* = 145) (**b**) stratified by development of PIC (NoPIC vs. PIC). CRP: C-reactive protein, PIC: postoperative infectious complications, POD: postoperative day, ns: no significance, *: *p* = 0.01, **: *p* ≤ 0.01.

**Figure 3 jcm-11-05654-f003:**
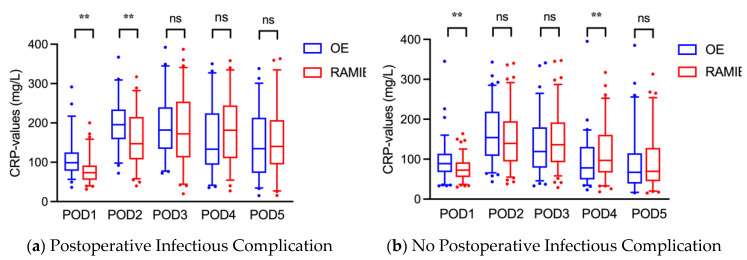
Postoperative CRP-values of patients with PIC (**a**) or without PIC (**b**) stratified by surgical approach. CRP: C-reactive protein, OE: open esophagectomy, POD: postoperative day, RAMIE: robot-assisted minimally invasive esophagectomy, ns: no significance, **: *p* ≤ 0.01.

**Table 1 jcm-11-05654-t001:** Baseline characteristics of patients undergoing RAMIE (*n* = 160) or OE (*n* = 145) stratified by surgical approach. Data are presented as mean ± SD, range or relative frequencies. Continuous variables were tested using ^a^ Students’ *t*-test (normally distributed) and ^b^ Mann-Whitney-U test (non-normally distributed). ASA: American Society of Anaesthesiologists; BMI: body mass index, OE: open esophagectomy, PIC: postoperative infectious complications; POD: postoperative day; RAMIE: Robot-assisted minimally-invasive esophagectomy, SD: Standard deviation. *p*-values < 0.05 were considered statistically significant and are indicated in bold.

	RAMIE (*n* = 160)	OE (*n* = 145)	*p*-Value
Sex (male [*n*, %])	131 [81.9]	116 [80.0]	0.63 ^b^
Age (years [mean ± SD])	64.0 ± 9.3	65.2 ± 10.3	0.29 ^a^
BMI (kg/m^2^, [mean ± SD])	26.6 ± 4.8	26.6 ± 5.2	0.97 ^a^
Smoking (yes [*n*, %])	57 [35.6]	45 [31.0]	0.70 ^b^
ASA ≥ 3 (yes [*n*, %])	107 [66.9]	96 [66.2]	0.90 ^b^
Entity [*n*, %]			0.05 ^b^
Adenocarcinoma	141 [88.1]	113 [66.2]	
Squamous cell carcinoma	15 [9.4]	23 [15.9]	
Other	4 [2.5]	9 [6.2]	
Procedure [*n*, %]			<0.001 ^a^
Ivor-Lewis	146 [88.1]	108 [75.5]	
Transhiatal oesophagectomy	5 [3.1]	29 [20.0]	
McKeown	9 [5.6]	8 [5.5]	
Length of surgery (min, [mean ± SD])	329.3 ± 74.1	293.0 ± 89.5	<0.010 ^a^
Neoadjuvant treatment [*n*, %]			
Radiotherapy	44 [27.5]	25 [17.2]	0.03 ^b^
Chemotherapy	123 [76.9]	86 [59.3]	<0.01 ^b^
Adjuvant treatment [*n*, %]			
Radiotherapy	5 [3.1]	10 [7.2]	0.11 ^b^
Chemotherapy	65 [42.2]	46 [38.7]	0.55 ^b^
PIC (yes [*n*, %])	64 (40.0)	59 (40.7)	0.09 ^b^
Length of hospital stay (days, [median, range])	20.5 (4–112)	19 (1–171)	0.59 ^b^

**Table 2 jcm-11-05654-t002:** Baseline and perioperative characteristics of patients undergoing RAMIE (*n* = 160) or OE (*n* = 145) stratified by PIC within 30 days after surgery. Data are presented as mean ± SD, range or relative frequencies. Continuous variables were tested using ^a^ Students’ *t*-test (normally distributed) and ^b^ Mann-Whitney-U test (non-normally distributed), while categorical variables were compared using ^c^ Chi-Square. ASA: American Society of Anaesthesiologists; BMI: body mass index, OE: open esophagectomy, PIC: postoperative infectious complications; POD: postoperative day; RAMIE: Robot-assisted minimally-invasive esophagectomy, SD: Standard deviation. *p*-values < 0.05 were considered statistically significant and are indicated in bold.

Postoperative Infectious Complication until POD 30			
	NoPIC (*n* = 182)	PIC (*n* = 123)	*p*-Value
Sex (male [*n*, %])	149 [48.9]	98 [39.7]	0.63 ^c^
Age (years [mean ± SD])	64.4 ± 9.9	64.8 ± 9.7	0.73 ^a^
BMI (kg/m^2^, [mean ± SD])	26.2 ± 4.8	27.3 ± 5.3	0.08 ^a^
Smoking (yes [*n*, %])	57 [18.7]	45 [14.8]	0.15 ^c^
ASA ≥ 3 (yes [*n*, %])	126 [41.3]	77 [25.3]	0.23 ^c^
Entity [*n*, %]			0.56 ^c^
Adenocarcinoma	155 [50.8]	99 [32.5]	
Squamous cell carcinoma	20 [6.6]	18 [5.9]	
Other	7 [2.3]	6 [2.0]	
Surgical approach [*n*, %]			
RAMIE	96 [31.8]	64 [21.0]	0.90 ^c^
OE	86 [28.2]	59 [19.3]	
Procedure [*n*, %]			0.83 ^c^
Ivor-Lewis	153 [50.2]	101 [33.1]	
Transhiatal oesophagectomy	20 [6.6]	14 [4.6]	
McKeown	9 [3.0]	8 [2.6]	
Length of surgery (min, [mean ± SD])	310.4 ± 81.2	314.7 ± 87.3	0.66 ^c^
Neoadjuvant treatment [*n*, %]			
Radiotherapy	38 [12.5]	31 [10.2]	0.38 ^c^
Chemotherapy	129 [42.3]	80 [26.2]	0.28 ^c^
Adjuvant treatment [*n*, %]			
Radiotherapy	11 [3.7]	4 [1.4]	0.28 ^c^
Chemotherapy	80 [29.3]	31 [11.4]	<0.001 ^c^
Length of hospital stay (days, [median, range])	14 (6–58)	39 (1–171)	<0.001 ^b^

**Table 3 jcm-11-05654-t003:** Postoperative CRP- and PCT-values of all patients included stratified by the development of PIC (NoPIC vs. PIC). CRP: C-reactive protein, PCT: Procalcitonin, PIC: postoperative infectious complications, POD: postoperative day. Continuous variables were tested using the Students’ *t*-test (normally distributed). *p*-values < 0.05 were considered statistically significant and are indicated in bold.

CRP (mg/L)	NoPIC (*n* = 182)	PIC (*n* = 123)	*p*-Value	PCT (mg/L)	NoPIC (*n* = 182)	PIC (*n* = 123)	*p*-Value
POD 1	84.5 ± 38.4	93.5 ± 42.7	0.07	POD 1	1.4 ± 2.9	2.1 ± 5.2	0.24
POD 2	155.5 ± 68.0	177.5 ± 69.1	<0.01	POD 2	1.1 ± 1.7	2.8 ± 11.9	0.23
POD 3	141.1 ± 70.5	186.3 ± 84.1	<0.01	POD 3	0.9 ± 1.2	2.4 ± 5.8	0.06
POD 4	107.0 ± 65.2	170.0 ± 84.6	<0.01	POD 4	1.2 ± 2.6	2.9 ± 10.1	0.26
POD 5	81.9 ± 70.5	150.7 ± 86.0	<0.01	POD 5	1.0 ± 2.2	3.6 ± 11.4	0.12
POD 2-1	70.9 ± 50.9	87.3 ± 57.8	0.02				
POD 3-2	−15.3 ± 47.4	12.7 ± 52.9	<0.01				
POD 4-3	−37.9 ± 40.9	−23.4 ± 50.0	0.02				

**Table 4 jcm-11-05654-t004:** Univariate logistic regression of CRP-levels depending on postoperative infectious complications and type of procedure. AUC and ROC-analysis to evaluate the prediction of a PIC. Best cut-off means the highest Youden’s index.

		Odds Ratio (95% CI)	*p*-Value	Cut-Off	AUC	NPV	PPV
**RAMIE**	POD 4	1.01 (1.01–1.02)	**<0.01**	168.0	0.73	73.9	65.2
	POD 5	1.01 (1.00–1.01)	**<0.01**	70	0.70	83.7	59.3
**OE**	POD 4	1.01 (1.01–1.02)	**<0.01**	88.3	0.74	80.4	58.9
	POD 5	1.01 (1.00–1.01)	**<0.01**	142.0	0.73	71.1	78.6

AUC: Area under the curve; CRP: C-reactive protein (mg/L); NPV: negative predictive value; POD: postoperative day; PPV: positive predictive value; ROC: receiver operating characteristic (ROC). *p*-values < 0.05 were considered statistically significant and are indicated in bold.

## Data Availability

All data included are available in this paper.

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
