# Peer review of "C-Reactive Protein as Predictor for Infectious Complications after Robotic and Open Esophagectomies"

_jcm, 2022, doi:10.3390/jcm11195654_

Round 1

Reviewer 1 Report

Dear Authors:

The manuscript by Richter et al. has demonstrated Post-operative CRP is a good negative predictor for PIC, after both RAMIE and OE. I have just a few suggestions.

1. Some background information and references are missing:

In introduction, page 1, Robotic approach is developed and used in many other fields, for example, breast cancer and reconstruction surgery. (please cite: 1. Efficacy of da Vinci robot-assisted lymph node surgery than conventional axillary lymph node dissection in breast cancer - A comparative study. Int J Med Robot. doi: 10.1002/rcs.2307.  2. Robot-Assisted Minimally Invasive Breast Surgery: Recent Evidence with Comparative Clinical Outcomes. J Clin Med. doi: 10.3390/jcm11071827.)

2. If it is possible, please explain or add a figure to show the potential mechanism of CRP to predict infectious complications

Best,

Author Response

Dear Reviewer 1,

please find enclosed our revised manuscript formerly entitled: C-reactive protein as an early predictor for infectious complications after robotic esophagectomy.

We thank you and the reviewers for the helpful suggestions and for the opportunity to revise our manuscript and kindly ask you to reconsider our article for publication in the Special Issue on “Postoperative Complications of Esophageal Cancer” in the Journal of Clinical Medicine.

According to the reviewers’ comments, we have now prepared a revised version of the manuscript that you will find enclosed with this letter. All changes have been highlighted in yellow for clarity.

We hope that the revised manuscript, together with this point-by-point discussion, will now address all concerns and render it suitable for publication in the Journal of Clinical Medicine.

Sincerely yours,

Florian Richter

Referee 1

Q1: Some background information and references are missing:

In introduction, page 1, Robotic approach is developed and used in many other fields, for example, breast cancer and reconstruction surgery. (please cite: 1. Efficacy of da Vinci robot-assisted lymph node surgery than conventional axillary lymph node dissection in breast cancer - A comparative study. Int J Med Robot. doi: 10.1002/rcs.2307.  2. Robot-Assisted Minimally Invasive Breast Surgery: Recent Evidence with Comparative Clinical Outcomes. J Clin Med. doi: 10.3390/jcm11071827.)

We thank the reviewer for these interesting references and have rewritten the section accordingly. The corresponding publications have been inserted.

Old:

RAMIE is proven to significantly influence the health-related quality of life (HRQoL) and long-term survival of the patients post esophagectomy [7].

New:

RAMIE is proven to significantly influence the health-related quality of life (HRQoL) and long-term survival of the patients post esophagectomy [5]. Robotic procedures have as well found their way into many other surgical fields, i.e. gynecological, hepatic, colorectal or reconstruction surgery [6-11].

Q2: If it is possible, please explain or add a figure to show the potential mechanism of CRP to predict infectious complications.

CRP is an acute phase protein which increases during the course of systemic inflammatory reactions, including both infectious complications and severe surgical trauma. However, it is difficult to differentiate the origin of the CRP-increase. Conducting the research for this paper, we therefore aimed to determine cut-off values and time points for CRP-increase, predicting a postoperative infectious complications beyond the expected postoperative CRP-increase. According to our data, patients with postoperative complications, including both surgical and non-surgical complications, presented with higher CRP-values starting from POD 2 and onwards. We further identified a slower and later CRP-increase in patients undergoing open esophagectomy compared to patients after robotic esophagectomy. We think that Figure 2 illustrates the differences in CRP-increases after the different procedures. Unfortunately, we are not able to provide a figure about the pathophysiological mechanisms underlying postoperative CRP-increases as this would be beyond the focus of this article. For clarity, we have added a references and modified the respective paragraph accordingly.

Old:

In addition to the clinical symptoms, white blood cell (WBC) count and the level of C-reactive protein (CRP) are crucial for the early detection of PIC. In almost all surgical procedures, an increase in WBC, CRP, and procalcitonin (PCT) can be observed within the first post-operative day (POD) itself [17]. In the absence of any PIC, the infectious values reach their maximum on the second or third POD and normalize in the course of the first post-operative week. In the presence of PIC the infection levels are usually higher initially and do not significantly drop during the post-operative course [18].

New:

Yet, both surgical trauma and PIC cause a systemic inflammatory infection, resulting in an increase in white blood cell (WBC) count, CRP, and procalcitonin (PCT) within the first post-operative day (POD) [21]. However, CRP and other inflammatory markers may not exactly determine the origin of the systemic inflammatory reaction. In the absence of any PIC, the infectious values reach their maximum on the second or third POD and normalize in the course of the first post-operative week [22]. Hence, WBC-count and the level of C-reactive protein (CRP) are crucial for the early detection of PIC in addition to the clinical symptoms.

Old:

Attempts have been made to correlate the predictability of PIC with CRP- and PCT-serum levels [19]. In course of the Enhanced Recovery After Surgery (ERAS)-program, CRP cut-off values are already taken into account when planning the early discharge of patients after major abdominal or colorectal surgery (Enhanced Recovery After Surgery) [20,21].

New:

Attempts have been made to correlate the predictability of PIC with CRP- and PCT-serum levels [24]. In course of the Enhanced Recovery After Surgery (ERAS)-program, CRP cut-off values are already taken into account when planning the early discharge of patients after major abdominal or colorectal surgery (Enhanced Recovery After Surgery) [25,26]. However, no exact cut-off values have been determined yet.

Reviewer 2 Report

The manuscript by Richter et al explored C-reactive protein as an early predictor for infectious complications after robotic esophagectomy. Overall, this topic is interesting. However, the analysis of post-operative infectious complications (PIC) is a bit superficial and broad. I would suggest they stratify different types of PIC and perform subgroup analysis.

1.     They should clearly define “post-operative infectious complications (PIC)”. Currently, they only explain which complications are included. But they didn’t illustrate the timeline. For example, infection occurred on POD2 and POD12 may have different infectious profiles.

2.     They should include the subgroup analysis based on the type of infection. Different infections have different patterns of CRP increase.

3.     What’s the role of CRP in predicting different types of infections?

4.     The PIC rates of RAMIE and OE are comparable. But the RAMIE had later and sometimes higher CRP level than OE. Any comments?

5.     The title is “C-reactive protein as an early predictor for infectious complications after robotic esophagectomy”. But the study focused on both RAMIE and OE. Please consider revising the title.

6.     “mg/l” should be “mg/L”.

7.      The title of Table 3 is incorrect.

Author Response

Dear Reviewer 2,

please find enclosed our revised manuscript formerly entitled: C-reactive protein as an early predictor for infectious complications after robotic esophagectomy.

We thank you and the reviewers for the helpful suggestions and for the opportunity to revise our manuscript and kindly ask you to reconsider our article for publication in the Special Issue on “Postoperative Complications of Esophageal Cancer” in the Journal of Clinical Medicine.

According to the reviewers’ comments, we have now prepared a revised version of the manuscript that you will find enclosed with this letter. All changes have been highlighted in yellow for clarity.

We hope that the revised manuscript, together with this point-by-point discussion, will now address all concerns and render it suitable for publication in the Journal of Clinical Medicine.

Sincerely yours,

Florian Richter

Referee 2

Q1: They should clearly define “post-operative infectious complications (PIC)”. Currently, they only explain which complications are included. But they didn’t illustrate the timeline. For example, infection occurred on POD2 and POD12 may have different infectious profiles.

We thank the reviewer for this comment and excuse for being unprecise about the time line of PIC. For clarity, we have added to following paragraph.

New:

In addition, we compared the time of onset of surgical (sPIC) and nonsurgical PIC (nsPIC) and the level of CRP at POD 4 for the different approaches.

NsPIC occurred in median at POD 4 (0-28) after RAMIE, and at POD 5.5 (0-18) after OE (p=0.55). CRP-values at POD 4 were 155.4 + 67.5 mg/L and 140.1 + 75.1 mg/L after RAMIE and OE, respectively (p=0.71).

sPIC occurred in median at POD 9 (1-30) and POD 7 (1-32) and CRP values at POD 4 were 193.3 + 85.6 mg/L and 161.0 + 84.9 mg/L (p=0.09), after RAMIE and OE, respectively. For both, RAMIE and OE, the differences in occurrence between sPIC and nsPIC as well as CRP levels at POD 4 were not statistically significant (p=0.37 and p=0.12).

Q2:They should include the subgroup analysis based on the type of infection. Different infections have different patterns of CRP increase.

We agree with the reviewer. We have hence differentiated between surgical postoperative complications (sPIC) and non-surgical postoperative complications (nsPIC) and performed an additional analysis comparing both subgroups.

Old:

All AL were detected through endoscopy, supported by a computed tomography, if necessary.

New:

All AL were detected through endoscopy, supported by a computed tomography, if necessary. For further evaluation, PIC were differentiated into surgical (sPIC) and nonsurgical (nsPIC). The former included leakage and necrosis of the gastric tube, the latter pulmonary complications.

Q3: What’s the role of CRP in predicting different types of infections?

CRP is an acute phase protein which increases during the course of systemic inflammatory reactions, including both infectious complications and severe surgical trauma. However, it is difficult to differentiate the origin of the CRP-increase. There are parameters considered to be more precise regarding postoperative infectious complications having a non-surgical origin than CRP. One of these parameters is PCT, which is generally considered to be more precise. However, our analysis regarding the analysis of PCT did not reveal any differences between the cohorts and was hence not considered to be helpful. Other parameteres, i.e. IL6, were not analyzed systematically in the patients included in this study. We therefore cannot report on these parameters.

However, by laboratory results alone, it is difficult to determine the origin of the inflammation. The laboratory results always need to be taken into consideration in combination with the clinical course. The aim of our study was mainly to define cut-off values which may predict an uncomplicated postoperative course. We have modified and added the following paragraphs accordingly

Old:

In addition to the clinical symptoms, white blood cell (WBC) count and the level of C-reactive protein (CRP) are crucial for the early detection of PIC. In almost all surgical procedures, an increase in WBC, CRP, and procalcitonin (PCT) can be observed within the first post-operative day (POD) itself [17]. In the absence of any PIC, the infectious values reach their maximum on the second or third POD and normalize in the course of the first post-operative week. In the presence of PIC the infection levels are usually higher initially and do not significantly drop during the post-operative course [18].

New:

Yet, both surgical trauma and PIC cause a systemic inflammatory infection, resulting in an increase in white blood cell (WBC) count, CRP, and procalcitonin (PCT) within the first post-operative day (POD) [21]. However, CRP and other inflammatory markers may not exactly determine the origin of the systemic inflammatory reaction. In the absence of any PIC, the infectious values reach their maximum on the second or third POD and normalize in the course of the first post-operative week [22]. Hence, WBC-count and the level of C-reactive protein (CRP) are crucial for the early detection of PIC in addition to the clinical symptoms.

Old:

Attempts have been made to correlate the predictability of PIC with CRP- and PCT-serum levels [19]. In course of the Enhanced Recovery After Surgery (ERAS)-program, CRP cut-off values are already taken into account when planning the early discharge of patients after major abdominal or colorectal surgery (Enhanced Recovery After Surgery) [20,21].

New:

Attempts have been made to correlate the predictability of PIC with CRP- and PCT-serum levels [24]. In course of the Enhanced Recovery After Surgery (ERAS)-program, CRP cut-off values are already taken into account when planning the early discharge of patients after major abdominal or colorectal surgery (Enhanced Recovery After Surgery) [25,26]. However, no exact cut-off value have been determined yet.

New:

Non-surgical complications may also impact reconvalescence seriously. The intention of our study was to calculate cut-off values to predict an uncomplicated course. Therefore, all PIC, sPIC as well as nsPIC, were combined. It is possible that a different steep and high increase of CRP in sPIC and nsPIC influence the predictive power. However, no significant difference could be found in our study for both forms of PIC with respect to time of onset as well as for the level of CRP at POD4.

Q4: The PIC rates of RAMIE and OE are comparable. But the RAMIE had later and sometimes higher CRP level than OE. Any comments?

We thank the reviewer for this interesting question. We attributed the later CRP-increase in RAMIE-patients to the different and overall smaller surgical trauma causing a different pattern of CRP-kinetics. Yet, we would like to point out that we included RAMIE-patients from the beginning of RAMIE-procedures in our Department, including longer surgical procedures and potentially comparably larger traumas potentially causing higher CRP-values. Even though CRP-values showed a later peak after RAMIE compared to OE, both curves are comparable.

Old:

Within the PIC-group, CRP-values in RAMIE-patients were significantly lower at POD 1 and 2, but higher at POD 4, compared to that of OE-patients due to a slower increase and a later peak (Figure 3).

New:

Within the PIC-group, CRP-values in RAMIE-patients were significantly lower at POD 1 and 2, but higher at POD 4, compared to that of OE-patients due to a slower increase and a later peak after RAMIE, yet without being significantly different (Figure 3).

Old:

Further, we found this delay to be independent of the occurrence of PIC. In case of PIC-development, post-operative CRP-levels differed from POD 2 onward, compared to patients without PIC, regardless of the type of procedure.

New:

Further, we found this delay to be independent of the occurrence of PIC. In case of PIC-development, post-operative CRP-levels differed from POD 2 onward, compared to patients without PIC, regardless of the type of procedure. We attributed the later CRP-peak after RAMIE to different CRP-kinetics after RAMIE.

Q5: The title is “C-reactive protein as an early predictor for infectious complications after robotic esophagectomy”. But the study focused on both RAMIE and OE. Please consider revising the title.

We thank the reviewer for discussing the titel of the manuscript. We agree with the reviewer and have adapted the titel of the manuscript.

Old:

C-reactive protein as an early predictor for infectious complications after robotic esophagectomy

New:

C-reactive protein as predictor for infectious complications after robotic and open esophagectomies

Q6:“mg/l” should be “mg/L”.

We thank the reviewer for this remarque and adapted the manuscript accordingly.

Old:

They also identified a good NUn score-based predictability for major complications on POD 4 with CRP-values of 219 ± 95 mg/l.

New:

They also identified a good NUn score-based predictability for major complications on POD 4 with CRP-values of 219 ± 95 mg/L.

Old:

Findlay et al. had previously reported on a predictive CRP-value of 156 mg/l using the NUn-score, focusing on results on POD 4 [52].

New:

Findlay et al. had previously reported on a predictive CRP-value of 156 mg/L using the NUn-score, focusing on results on POD 4 [52].

Q7: The title of Table 3 is incorrect.

We thank the reviewer for noticing our mistake. The heading of Table 3 has been corrected accordingly.

Old:

Table 3. Postoperative CRP- and PCT-values of patients of patients undergoing RAMIE (n = 160) or OE (n = 145) stratified by development of PIC (NonPIC vs. PIC).

New:

Table 3. Postoperative CRP- and PCT-values of all patients included stratified by development of PIC (NonPIC vs. PIC).

Reviewer 3 Report

Please make sure no grammatical or technical errors before being concsidered for publication. For example in the abstract, respec-tively.

 PIC were noted in 170 patients on postoperative day (POD) 30 and 127 on POD 10, respec-tively.

Author Response

Dear Dr. Tanaka,

please find enclosed our revised manuscript formerly entitled: C-reactive protein as an early predictor for infectious complications after robotic esophagectomy.

We thank you and the reviewers for the helpful suggestions and for the opportunity to revise our manuscript and kindly ask you to reconsider our article for publication in the Special Issue on “Postoperative Complications of Esophageal Cancer” in the Journal of Clinical Medicine.

According to the reviewers’ comments, we have now prepared a revised version of the manuscript that you will find enclosed with this letter. All changes have been highlighted in yellow for clarity.

We hope that the revised manuscript, together with this point-by-point discussion, will now address all concerns and render it suitable for publication in the Journal of Clinical Medicine.

Sincerely yours,

Florian Richter

Referee 3

Q1: PIC were noted in 170 patients on postoperative day (POD) 30 and 127 on POD 10, respec-tively.

We thank the reviewer for pointing out this mistake. We have again checked the manuscript for grammatical errors and corrected mistakes.

Old:

PIC were noted in 170 patients on postoperative day (POD) 30 and 127 on POD 10, respectively

New:

PIC were noted in 170 patients on postoperative day (POD) 30 and 127 on POD 10.

Round 2

Reviewer 1 Report

Strongly suggest for publication.

Reviewer 2 Report

The authors have addressed my concerns.